# Neutrophil metabolomics in severe COVID-19 reveal GAPDH as a suppressor of neutrophil extracellular trap formation

Yafeng Li[1], Jessica S. Hook [2], Qing Ding[1], Xue Xiao[2,3], Stephen S. Chung[4], Marcel Mettlen[5], Lin Xu [2,3], Jessica G. Moreland [2,6] & Michalis Agathocleous [1,2] ✉

Severe COVID-19 is characterized by an increase in the number and changes in the function of innate immune cells including neutrophils. However, it is not known how the metabolome of immune cells changes in patients with COVID-19. To address these questions, we analyzed the metabolome of neutrophils from patients with severe or mild COVID-19 and healthy controls. We identified widespread dysregulation of neutrophil metabolism with disease progression including in amino acid, redox, and central carbon metabolism. Metabolic changes in neutrophils from patients with severe COVID-19 were consistent with reduced activity of the glycolytic enzyme GAPDH. Inhibition of GAPDH blocked glycolysis and promoted pentose phosphate pathway activity but blunted the neutrophil respiratory burst. Inhibition of GAPDH was sufficient to cause neutrophil extracellular trap (NET) formation which required neutrophil elastase activity. GAPDH inhibition increased neutrophil pH, and blocking this increase prevented cell death and NET formation. These findings indicate that neutrophils in severe COVID-19 have an aberrant metabolism which can contribute to their dysfunction. Our work also shows that NET formation, a pathogenic feature of many inflammatory diseases, is actively suppressed in neutrophils by a cell-intrinsic mechanism controlled by GAPDH.

Severe COVID-19 is driven by pathogenic hyperinflammation, which includes increased neutrophils in the blood and lungs[1–9]. Neutrophils produce neutrophil extracellular traps (NETs), chromatin structures which propagate inflammation[10]. NET levels are increased in severe COVID-19 patient serum and lung samples[11–17], contribute to pathology[11,14,18,19] and can predict mortality[20]. Therefore, inhibition of NET formation may ameliorate severe disease. The mechanisms regulating NET formation have attracted considerable interest because NETs are thought to be pathogenic in many other diseases including sepsis[21], malaria[22], atherosclerosis[23], diabetes[24], stroke[25], coagulation/thrombosis[26,27], myocardial infarction[28], rheumatic diseases[29], and pancreatitis[30]. NET formation and the neutrophil respiratory burst are regulated by metabolic mechanisms downstream of extracellular inflammatory signals[31–40]. While many extracellular signals are sufficient to induce NET formation, it is not known if neutrophils possess cell-intrinsic,

[1]Children's Medical Center Research Institute, University of Texas Southwestern Medical Center, Dallas, TX, USA. [2]Department of Pediatrics, University of Texas Southwestern Medical Center, Dallas, TX, USA. [3]Quantitative Biomedical Research Center, Department of Population and Data Sciences, University of Texas Southwestern Medical Center, Dallas, TX, USA. [4]Department of Internal Medicine, Division of Hematology and Oncology, University of Texas Southwestern Medical Center, Dallas, TX, USA. [5]Department of Cell Biology, Quantitative Light Microscopy Core, University of Texas Southwestern Medical Center, Dallas, TX, USA. [6]Department of Microbiology, University of Texas Southwestern Medical Center, Dallas, TX, USA. ✉e-mail: michail.agathokleous@utsouthwestern.edu

NET-suppressing mechanisms whose disruption is sufficient to induce NET formation.

The metabolism of immune cells plays a key role in their function[41]. Because immune dysregulation is a main feature of severe COVID-19 and other inflammatory diseases, it is important to understand how the metabolism of immune cells changes with disease progression. However, a comprehensive metabolomic characterization of neutrophils or other immune cells in COVID-19 patients has not been reported. Therefore, it is not known how the metabolome of immune cells changes in COVID-19 or how such changes can regulate immune cell function. In this work, we perform a comprehensive metabolomics analysis of neutrophils in COVID-19 patients. We show widespread metabolic changes in neutrophils from severe COVID-19 patients as compared to neutrophils from mild COVID-19 patients or healthy controls. GAPDH activity was inhibited in neutrophils from severe patients. We use these results as a starting point to identify GAPDH as a neutrophil-intrinsic NET-suppressor mechanism and to investigate the molecular mechanisms by which GAPDH inhibition regulates neutrophil function.

## Results

### The neutrophil metabolome in COVID-19
To understand how the metabolome of neutrophils changes in COVID-19 we performed metabolomics from polymorphonuclear leukocytes, more than 95% of which are neutrophils (Fig. S1a)[17], isolated from patients with severe/critical COVID-19 ($n = 26$, hereafter referred to as 'severe'), mild/moderate COVID-19 ($n = 30$, hereafter referred to as 'mild'), or healthy controls ($n = 19$) (Fig. S1b). Patients in the severe COVID-19 group had acute respiratory distress syndrome and were being treated in the intensive care unit. 77% of them received mechanical ventilation support. Of patients in the mild COVID-19 group, 52% had symptoms of pneumonia and the remainder had milder symptoms typical of COVID-19. Neutrophil metabolites were extracted and analyzed using liquid chromatography-mass spectrometry (LC-MS). A total of 287 metabolites were quantified. Unsupervised principal component analysis (PCA) of the raw data showed tightly clustered quality control (QC) samples (Fig. S1c), consistent with stable LC-MS performance without significant systematic drift. Neutrophils from severe COVID-19 patient samples were metabolically distinct from healthy controls (Fig. 1a). Neutrophil samples from mild COVID-19 patients clustered between samples from severe COVID-19 patients and healthy controls, consistent with a range of disease severity in this group (Fig. 1a). The levels of 136 metabolites significantly differed between severe COVID-19 and controls, 111 metabolites differed between mild COVID-19 and controls, and 56 metabolites differed between severe and mild COVID-19 (Fig. 1b–d, Supplementary Datasets S1–S2). Pathway enrichment analysis revealed that differentially abundant metabolites in severe as compared to mild COVID-19 or healthy controls belonged to a small number of metabolic pathways, including central carbon metabolism, histidine/β-alanine metabolism, and lipid synthesis (Fig. 1e–g). Differentially enriched metabolites fell into diverse metabolite classes (Fig. 1h). Untargeted metabolomics detected 1337 features, 918 of which significantly differed between treatments (Fig. S1d). Features enriched in severe COVID-19 neutrophils included tentatively identified lipids and dipeptides (Fig. S1d). Medications such as propofol, azithromycin and its metabolite azithralosamine were also observed (Fig. S1d–g), indicating they can enter and may concentrate in neutrophils. To map relationships between metabolites and test how they change with disease progression we performed correlation-based networking analysis (Figs. 1i–k, S1h). Compared with healthy controls or mild COVID-19 patients, severe COVID-19 was associated with the emergence of distinct metabolite hubs in neutrophils, including a hub consisting of glycolytic metabolites suggesting coordinated changes in neutrophil glycolysis (Fig. 1k). The levels of many amino acids were

correlated with each other in neutrophils from the controls but were uncorrelated in neutrophils from patients, suggesting global disruption of neutrophil amino acid metabolism (Fig. S1i). After controlling for potential confounding variables of age, gender, and sample collection time, most metabolic differences between groups remained including 121 metabolites which significantly differed between severe COVID-19 and controls, 99 metabolites between mild COVID-19 and controls, and 75 metabolites between severe and mild COVID-19 (Supplementary Datasets S3–S4). Therefore, severe COVID-19 was accompanied by widespread metabolic rewiring in neutrophils.

### Dysregulation of neutrophil amino acid and redox metabolism in severe COVID-19
Depletion of several amino acids was observed in neutrophils in severe COVID-19 patients (Fig. S2a), while many lipids were elevated (Fig. S2b). A consistently altered metabolic pathway in severe COVID-19 was histidine/β-alanine metabolism (Fig. 1f, g). Histidine levels in patient neutrophils declined (Fig. S2c). β-alanine, which reacts with histidine to form the dipeptides carnosine and anserine, was significantly elevated in patient neutrophils (Fig. S2c). Neutrophils express high levels of the taurine/β-alanine importer *SLC6A6* (Fig. S2d) suggesting they concentrate β-alanine. Metabolites of uracil degradation, a source of β-alanine, were elevated in neutrophils of severe COVID-19 patients consistent with elevated β-alanine production (Fig. S2c). Carnosine was reduced and anserine was almost undetectable in severe patient neutrophils (Fig. S2c). Severe patient neutrophils had changes in levels of many redox-related metabolites. They were enriched in β-hydroxycarnitines (Fig. S3a), markers of mitochondrial impairment or hypoxia[42–44] and had a 10-fold reduction in thioproline (Fig. S3b), which can function as a cysteine-derived antioxidant[45]. Severe patient neutrophils had elevated markers of oxidative stress including ophthalmate (Fig. S3c), a marker of cysteine and glutathione depletion[46,47], and methylguanine (Fig. S3d) a marker of reactive nitrogen species-induced DNA damage[48]. A major antioxidant in human plasma is ascorbate[49]. In prior studies, ascorbate was undetectable in the plasma of severe COVID-19 patients[50,51]. To accurately quantify ascorbate concentrations, we used targeted LC-MS with spiked-in labeled ascorbate internal standards. Plasma ascorbate concentration modestly declined in mild COVID-19 patients, and significantly declined in severe COVID-19 patients (Fig. S3e). Ascorbate was detectable in almost all patient samples. Half of the severe patients had plasma ascorbate concentrations below 20 µM (Fig. S3e). Vitamin C supplementation in 8 patients was associated with normal ascorbate concentration (Fig. S3f). In neutrophils, ascorbate levels in mild patients declined as compared to controls, but levels in severe COVID-19 patients did not (Fig. S3g). Plasma ascorbate concentration did not correlate with neutrophil ascorbate levels but correlated with levels of threonate (Fig. S3h), a product of oxidized ascorbate, consistent with transport of oxidized ascorbate in neutrophils[52]. Therefore, neutrophils in COVID-19 patients were characterized by depletion of many amino acids, accumulation of β-alanine, anserine depletion, and several changes in metabolite levels indicative of redox stress.

### GAPDH inhibition in neutrophils of severe COVID-19 patients and the role of GAPDH in NET formation
Changes in central carbon metabolism including glycolysis and the pentose phosphate pathway were salient features of neutrophils in severe COVID-19 (Fig. 1f, k). The ratio of dihydroxyacetone phosphate:1,3-bisphosphoglycerate (DHAP:BPG) increased in severe COVID-19 neutrophils, consistent with a reduction in GAPDH activity (Fig. 2a, b). To understand whether GAPDH inhibition affects neutrophil function, we treated freshly isolated neutrophils from healthy donors with the GAPDH inhibitor heptelidic acid (HA), a natural metabolite of the fungus *T. koningii*[53]. *T. koningii* expresses an HA-resistant GAPDH[54], which can rescue the effects of HA on cell viability

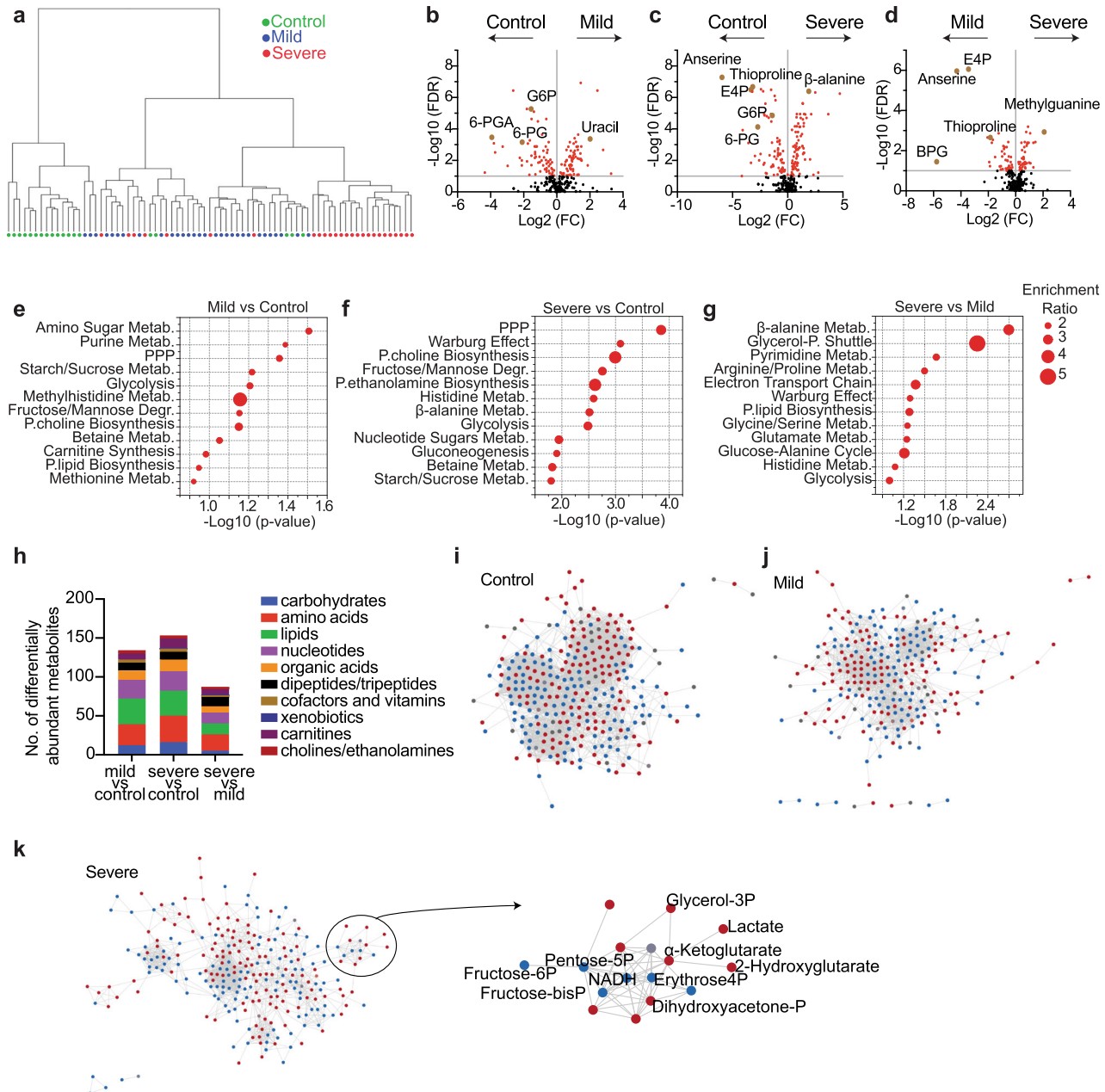

**Fig. 1 | Changes in the neutrophil metabolome in severe or mild patients with COVID-19 and healthy controls. a** Unsupervised hierarchical clustering of metabolomic analysis of neutrophils from control (healthy adults, *n* = 19), mild (*n* = 30), or severe (*n* = 26) COVID-19 patients. Clustering was performed using the Euclidean distance measure with Ward clustering algorithm. **b–d** Volcano plots comparing the neutrophil metabolome of control, mild, or severe COVID-19 patients. Each dot represents a metabolite. The names of some of the metabolites discussed in the Results are shown. G6P glucose 6-phosphate, 6-PG 6-phosphogluconate, 6-PGA 6-phosphogluconolactone, E4P erythrose 4-phosphate, BPG 1,3-bisphosphoglycerate. **e–h** Pathway enrichment analysis (**e–g**) and classification (**h**) of neutrophil metabolites that significantly changed in the indicated comparisons. **i–k** Network diagrams of metabolites in neutrophils from control, mild, or severe COVID-19 patients. A magnification of a new cluster of glycolytic and PPP metabolites in severe patients is shown in (**k**). Red dots represent elevated metabolites and blue dots depleted metabolites in neutrophils from severe COVID-19 patients compared with healthy controls. Statistical significance was assessed with two-tailed *t*-tests followed by multiple comparisons correction (**b–d**). In all figures, multiple comparisons correction was performed by controlling the false discovery rate at 5% using the method of Benjamini, Krieger, and Yekutieli for metabolomics analysis, or by using multiple comparisons tests as described in the methods.

and growth of mammalian cells[55], suggesting HA is a specific GAPDH inhibitor. GAPDH inhibition in neutrophils increased staining with a cell-impermeable DNA stain, as did the NET inducer phorbol 12-myristate 13-acetate (PMA) (Fig. 2c, d). To assess cell morphology after GAPDH inhibition and distinguish NET formation from other forms of cell death[34,56], we used time-lapse live-cell microscopy (Fig. S4a). HA treatment caused permeabilization of the cell membrane and nuclear expansion followed by changes in nuclear morphology and spilling of DNA outside the cell membrane (Movies 1–2 and Fig. 2e, f). The

morphological characteristics of HA-induced NET formation differed from those of PMA-induced NET formation, in which loss of nuclear lobular structure and nuclear expansion preceded cell membrane permeabilization (Movie 3 and Fig. 2e, f). This is consistent with findings that different stimuli induce NETs by different mechanisms[34]. Another known NET inducer, the calcium ionophore A23187, induced NETs characterized by cell membrane permeabilization followed by nuclear expansion, a process which resembled HA-induced NET formation more than PMA-induced NET formation (Movies 4–7 and

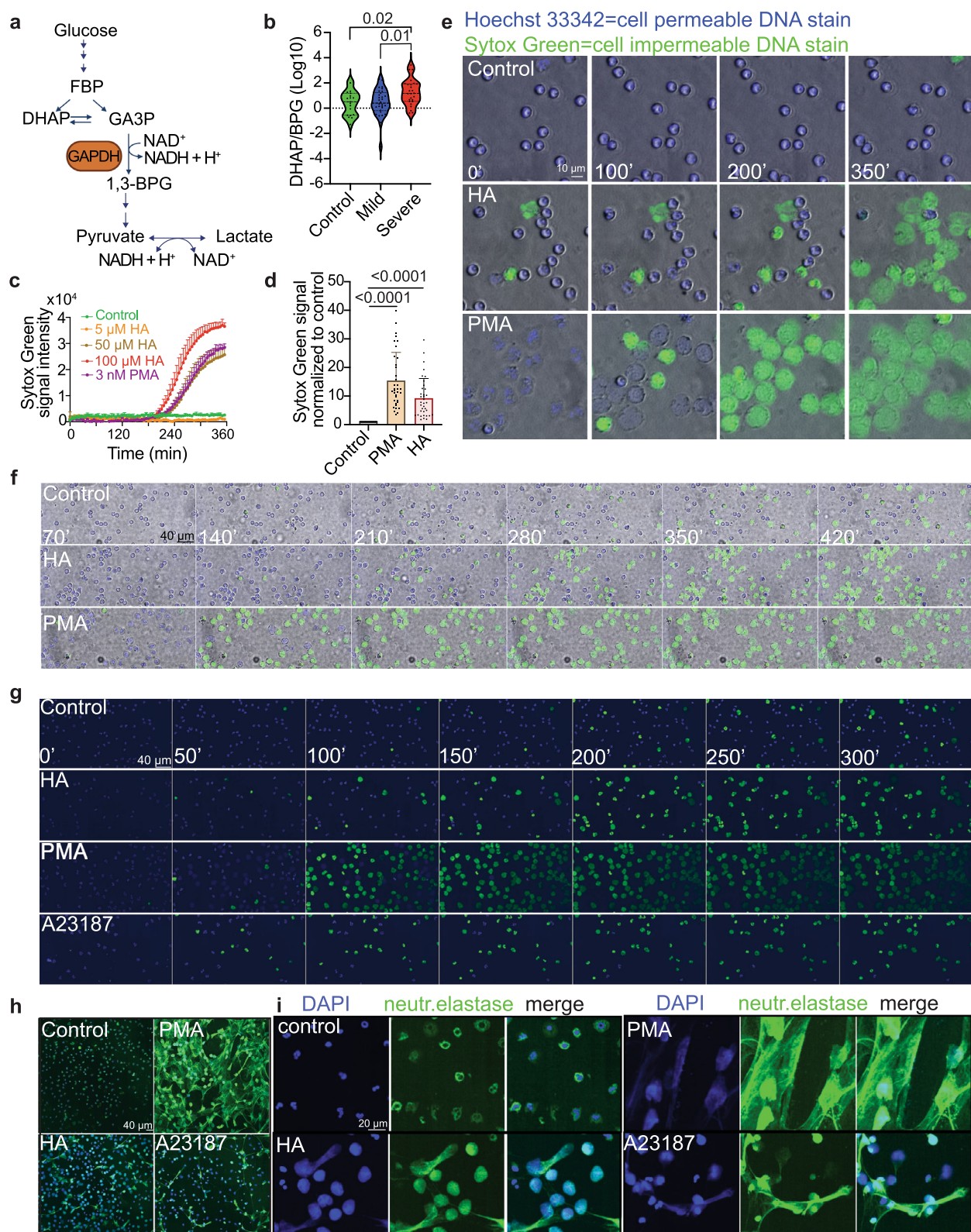

Fig. 2g). A23187 caused the appearance of extracellular Hoechst-positive speckles which were not present in the other two treatments (Movie 7). Treatment with HA but not with PMA or A23187 froze neutrophil nuclear motion prior to NET formation (Movies 1–7). HA-treated neutrophils showed nuclear neutrophil elastase staining comparable to that caused by A23187, and weaker than that caused by PMA (Fig. 2h, i). HA-treated neutrophils showed histone citrullination

staining which was weaker than that caused by A23187 (Fig. S4b), consistent with variable histone citrullination in response to different NET-inducers[34]. To test if a different GAPDH inhibitor also caused NET formation, we treated neutrophils with iodoacetate (IA). Iodoacetate induced NET formation with morphological attributes and kinetics identical to those induced by HA (Movies 8–10 and Fig. S4c). Thus, GAPDH inhibition in primary human neutrophils was sufficient to

**Fig. 2 | GAPDH inhibition promotes neutrophil extracellular trap formation.**
**a** Diagram of the GAPDH reaction. **b** The DHAP/BPG ratio in neutrophils from COVID-19 patients and healthy controls. Violin plot lines represent median and interquartile range. **c, d** Effect of HA or PMA on Sytox Green fluorescent signal intensity in primary human neutrophils. A representative trace (**c**) and quantitation of independent experiments (**d**) are shown. In (**d**) datapoints represent human donor neutrophil samples from $n = 18$ independent experiments, tested twice/ experiment, from a total of $n = 8$ human donors. Some of these experiments included other treatments described subsequently in the paper. HA Heptelidic acid, PMA Phorbol 12-myristate 13-acetate. **e, f** Time-lapse images of the effects of HA, PMA, or A23187 on NET formation. **e** shows magnified images of 4 timepoints, and **f, g** show larger fields of view at different timepoints. **e, f** were acquired with, and **g** without, the brightfield channel. Representative images from $n = 22$ independent experiments for HA, $n = 14$ for PMA treatment, and $n = 3$ for A23187 treatment. **h, i** Neutrophil elastase staining after HA, PMA, or A23187 treatment. Panel **i** shows a magnification from **h**. Representative images from four independent experiments. Data represent median and interquartile range (**b**), mean ± s.e.m. (**c**), mean ± st.dev. (**d**). Statistical significance was assessed with one-way ANOVA followed by multiple comparisons correction (**b**) or one-way ANOVA with Brown–Forsythe correction and Dunnett's T3 multiple comparisons test (**d**).

cause NET formation. NET formation induced by GAPDH inhibition shared some morphological characteristics with NET formation induced by PMA or A23187, and differed from them in other aspects, suggesting a distinct cell-intrinsic NET formation mechanism.

### The effects of TNF-α stimulation and GAPDH inhibition on glycolysis and the PPP

Severe COVID-19 is characterized by elevated inflammatory cytokines including TNF-α[57]. To understand how GAPDH inhibition impacts glycolysis and the PPP in primary human neutrophils in the presence or absence of inflammatory challenge, we performed stable isotope tracing in TNF-α-treated and HA-treated neutrophils using uniformly labeled glucose (U[13]C-glucose) and 1,2-[13]C labeled glucose. TNF-α increased glucose-derived labeling in almost all measured glycolytic intermediates, suggesting stimulation of glycolysis (Fig. 3a, b) in agreement with findings that inflammatory stimuli promote neutrophil glycolysis[38–40,58]. In U[13]C-glucose tracing, m + 6 labeling of fructose-1,6-bisphosphate (FBP) and fructose-6-phosphate (F6P) is derived from forward flow in glycolysis, and m + 3 labeling from backward flow of 1 labeled and 1 unlabeled triose phosphate through aldolase (Fig. 3a). The ratio of m + 6 to m + 3 labeled species for G6P, F6P and FBP increased after TNF-α stimulation (Fig. 3c) consistent with increased glycolysis. HA treatment decreased labeling in lower glycolysis, as expected from GAPDH inhibition (Fig. 3b). HA treatment decreased the m + 6/m + 3 ratio in G6P, F6P, and FBP (Fig. 3c), suggesting decreased forward relative to backward flux in upper glycolysis.

To assess relative flow of upper glycolysis and the oxidative pentose phosphate pathway (oxPPP), we used 1,2-[13]C-glucose[59], which produces m + 2 labeled lower glycolysis metabolites when metabolized via upper glycolysis and m + 1 labeled metabolites when metabolized via the oxPPP (Fig. 3d). TNF-α treatment decreased the m + 1/m + 2 ratio in DHAP, 2-/3-phosphoglycerate, pyruvate, and lactate (Figs. 3e, S5a–d) suggesting increased flow in glycolysis relative to the oxPPP. Pentose phosphates (PP, including ribose-5P, ribulose-5P and xylulose-5P) have 1 labeled carbon if they are derived from the oxPPP and 2 labeled carbons if they are derived from the non-oxPPP (Fig. 3d). TNF-α treatment decreased m + 1 and increased m + 2 and m + 4 pentose phosphate, suggesting decreased oxPPP relative to non-oxPPP activity (Fig. 3f). GAPDH inhibition caused significant changes in PPP metabolite labeling. In U[13]C-glucose labeling, HA treatment increased the proportion of partially labeled PP isotopologues (m + 2, m + 3, m + 4, Fig. S5e), which can arise from interconversions in the PPP, and the proportion of m + 4 sedoheptulose 7-P (S7P, Fig. S5f), which can arise from labeled erythrose 4-Phosphate (E4P) through transaldolase or aldolase in the non-oxidative PPP. This was consistent with increased PPP activity. In 1,2-[13]C-glucose tracing, HA treatment decreased m + 2 and m + 1 pyruvate (Fig. 3g) suggesting that GAPDH inhibition blocks contribution of both upper glycolysis and the oxPPP to lower glycolysis as expected. Cycling between the oxPPP and glycolysis decreases labeling as labeled carbons are lost in the 6-phosphogluconate (6-PG) dehydrogenase reaction (Fig. 3h). HA treatment decreased m + 2 and increased m + 0 6-phosphogluconolactone (6-PGA, Fig. S5g) and 6-phosphogluconate (6-PG, Fig. 3i). Because almost all carbons in 6-PGA and 6-PG derive from glucose in both control and HA treatment

(Fig. S5h, i), these results suggested that HA increased cycling through the oxPPP and glycolysis. HA treatment decreased F6P and G6P m + 2 labeling and increased unlabeled F6P and G6P (Fig. 3j, k). Because the contribution of U[13]C-glucose-derived carbon to F6P and G6P did not decrease after HA treatment (Fig. S5j, k), this result is consistent with increased cycling between the oxPPP and glycolysis. HA treatment also increased PP m + 2 labeling while total PP labeling from U[13]C-glucose did not change (Figs. 3l, S5l), suggesting increased interconversion in the non-oxidative PPP. Our data is consistent with prior data that GAPDH inhibition in cell lines promotes PPP activity[60–62]. GAPDH inhibition had similar effects on glycolysis and the PPP in the presence of TNF-α (Figs. 3b, c and S5e, f). Therefore TNF-α promoted glycolysis and HA treatment blocked glycolysis at the level of GAPDH and promoted activity of the PPP (Fig. 3m).

### GAPDH inhibition induces NETs independently of the oxPPP and NOX activity

Some pathways that induce NET formation require oxPPP/NADPH-dependent ROS production by NOX2 in the respiratory burst[31–35]. Because HA stimulated the oxPPP, we tested if HA-induced NET formation required the oxPPP or the respiratory burst. To induce the respiratory burst, HA-treated neutrophils were stimulated with TNF-α followed by N-formylmethionyl-leucyl-phenylalanine (fMLF). Surprisingly HA blocked respiratory burst-dependent ROS production (Fig. 4a, b). Because GAPDH inhibition blocks the metabolism of triose phosphates, we hypothesized that this would lead to accumulation of methylglyoxal, a reactive metabolite which forms non-enzymatically from DHAP (Fig. 4c). To test that, we measured neutrophil methylglyoxal levels. Methylglyoxal accumulated after HA treatment (Fig. 4d). Labeling with U[13]C-glucose showed that the source of most methylglyoxal in HA-treated neutrophils was glucose, consistent with methylglyoxal formation from glycolysis (Fig. 4e). Methylglyoxal treatment blocked respiratory burst-dependent ROS production (Fig. 4f–h) suggesting that GAPDH inhibition-induced methylglyoxal elevation can impair the respiratory burst. Blocking the oxPPP using a G6PD inhibitor (G6PDi)[32], or blocking NOX activity using diphenyleneiodonium (DPI)[33] suppressed PMA-induced NET formation but did not suppress HA-induced NET formation (Figs. 4i–k, S6 and Movies 11–16). DPI treatment accelerated HA-induced NET formation (Figs. 4j, k, S6d, e), consistent with data in mice that Nox2-deficiency promotes NET formation and uncontrolled sterile inflammation[63–66]. Therefore, GAPDH inhibition blocked the respiratory burst and induced NET formation which did not require the oxPPP or the respiratory burst.

### Metabolic mechanisms of GAPDH inhibition-induced NET formation

To understand how GAPDH inhibition changes neutrophil metabolism to regulate NET production, we performed metabolomics in HA-treated neutrophils. HA treatment increased levels of DHAP, FBP, and F6P and decreased levels of pyruvate and malate (Fig. 5a–e), consistent with a block of glycolysis at the level of GAPDH. HA treatment increased the NAD[+]/NADH ratio, consistent with loss of GAPDH activity (Fig. 5f), and changed the levels of several PPP metabolites (Fig. 5g–i).

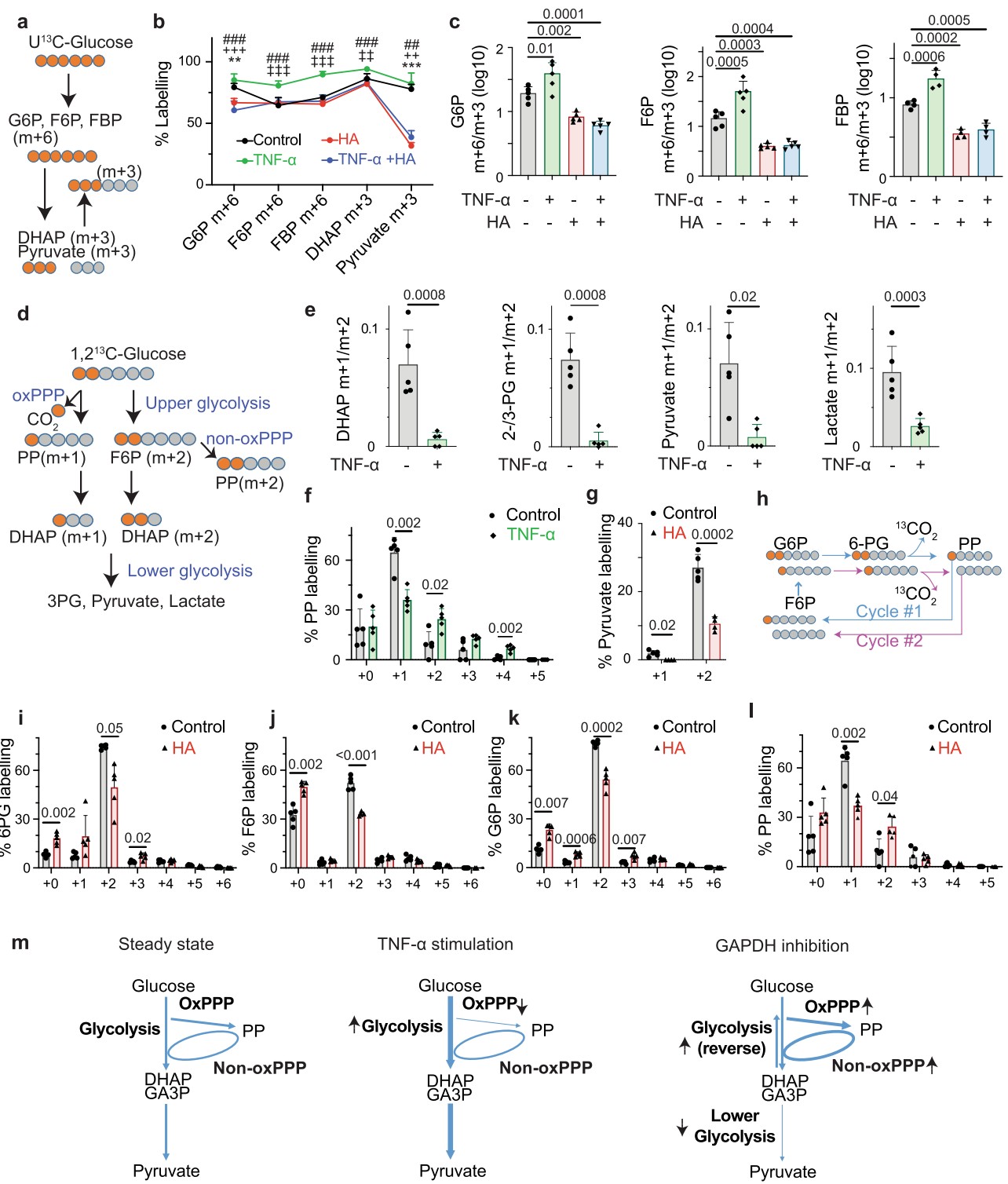

Some of the HA-induced metabolic changes overlapped with changes we observed in severe COVID-19 patients as compared to healthy controls (Fig. S7a). Other changes in severe COVID-19 neutrophils, such as in histidine/β-alanine pathway metabolites were not observed after HA treatment, suggesting multiple factors cause metabolic changes in severe COVID-19 neutrophils. To understand if NET formation after HA or iodoacetate treatment was associated with GAPDH inhibition, we examined the effects of two doses of HA or iodoacetate on metabolite levels. NET-inducing doses of HA or iodoacetate caused almost identical changes in glycolysis and the PPP (Fig. S7b−l). At NET-inducing doses, HA or iodoacetate highly elevated metabolites upstream of GAPDH and decreased metabolites downstream of GAPDH. A lower dose of HA which did not induce NETs had minimal effects on metabolites upstream and downstream of GAPDH, and a lower dose of iodoacetate which induced intermediate NET levels had intermediate effects on metabolites upstream and downstream of GAPDH (Fig. S7b−l). Therefore, the NET-inducing ability of HA or iodoacetate correlates with their inhibition of GAPDH activity. GAPDH inhibition in macrophages elevates mitochondrial ROS which promote inflammasome activation and pyroptosis[67], and mitochondrial ROS in neutrophils can induce NETs[68]. However, HA treatment did not elevate mitochondrial ROS or other ROS species and did not decrease the

**Fig. 3 | GAPDH inhibition or TNF-α stimulation affect neutrophil glycolysis and the pentose phosphate pathway. a** Flow of $U^{13}C$-glucose-derived carbons in glycolysis. Forward flow of glycolysis produces $m + 6$ hexose phosphates, and $m + 3$ DHAP and pyruvate. Backward flow through aldolase produces $m + 3$ hexose phosphates from one fully labeled (orange circles) and one fully unlabeled (gray circles) triose phosphate. **b, c** Effects of TNF-α and HA treatment on $U^{13}C$-glucose-derived labeling of glycolytic intermediates (**b**) and in +6 labeling relative to +3 labeling in upper glycolytic intermediates (**c**) in primary human neutrophils. **d** Source of $1,2^{13}C$-glucose-derived carbons in pentose phosphate pathway (PPP) and lower glycolysis. **e, f** Effects of TNF-α treatment on $1,2^{13}C$-glucose-derived $m + 1$ labeling (oxPPP-derived) relative to $m + 2$ labeling (glycolysis-derived) of lower glycolysis intermediates and pentose phosphates (PP). **g** Effects of HA treatment on $1,2^{13}C$-glucose-derived labeling of pyruvate. **h–l** Source of $1,2^{13}C$-glucose-derived carbons in PPP intermediates after cycling between oxPPP and glycolysis (**h**), and the effects of HA treatment (**i–l**). **m** Summary of the effects of TNF-α treatment or GAPDH inhibition on glycolysis and the PPP in primary human neutrophils. Data represent mean ± st.dev. In (**c–l**) $p$-values are shown on the graphs. In (**b**) * compares control versus HA, ‡ compares control versus TNF-α, + compares control versus TNF-α + HA, and # compares TNF-α versus TNF-α + HA and *$p < 0.05$, **$p < 0.01$, ***$p < 0.001$. $n = 5$ donors from 5 independent experiments. Statistical significance was assessed with one-way ANOVA with Holm–Sidak multiple comparisons correction (**b**), one-way ANOVA of log-transformed values with Holm–Sidak multiple comparisons correction (**c**), one-way ANOVA with Brown–Forsythe correction and Dunnett's T3 multiple comparisons test (**d**), two-tailed Mann–Whitney test (for DHAP, 3-PG, and pyruvate in **e**, and for $m + 1$ in **g**), two-tailed $t$-test of log-transformed values (for lactate in **e**), two-tailed $t$-test with Holm–Sidak multiple comparisons correction (for $m + 2$ in **g, i–l**), two-tailed Welch's $t$-test with Holm–Sidak multiple comparisons correction (for $m + 2$ in **i**). In panels **f** and **l**, and **g** and S5c, HA and TNF-α treatments were tested in parallel with the same control treatment.

GSH/GSSG ratio (Fig. S8a, b), suggesting it does not trigger NET formation by elevating ROS. Supplementation of neutrophils with pyruvate or lactate did not rescue HA-induced NET formation (Fig. S8c–e, Movies 17–22). Inhibition of flow through upper glycolysis by 2-deoxyglucose (2DG) delayed HA-induced NET formation (Fig. 5j, Movies 23–26). Consistent with inhibition of upper glycolysis, 2DG partially rescued the accumulation of upper glycolysis metabolites caused by GAPDH inhibition (Fig. 5k–m) but did not rescue the loss of lower glycolysis metabolites or $NAD^+$, NADH, ATP levels (Fig. 5n–r). These results suggest that GAPDH inhibition-induced NET formation is not caused by a general reduction in glycolysis rate, loss of ATP, or loss of glycolytic end products, but specifically by a block at the level of GAPDH. Therefore, metabolic flux through GAPDH suppresses NET formation.

## GAPDH inhibition causes neutrophil death and NET formation by increasing the pH

The GAPDH reaction is coupled with $H^+$ production (Fig. 2a). Consistent with a reduction in $H^+$ production, GAPDH inhibition increased the neutrophil intracellular pH (Fig. 6a). Decreasing the extracellular pH using HEPES buffer neutralized the increase in neutrophil pH caused by GAPDH inhibition (Fig. 6b) and completely rescued neutrophil death and NET formation (Fig. 6c, Movies 27–30). To test the involvement of changes in pH with an alternative approach, we inhibited $Na^+/H^+$ exchangers, which control intracellular pH by extruding $H^{+69}$. Neutrophils express the highest levels of NHE1 and NHE8 (*SLC9A1/SLC9A8*) among blood cells (Fig. S9a) and do not express, or express low levels of other NHE family members[70]. Cariporide inhibits NHE1 and NHE8[71,72]. At doses that normalized the increase in pH caused by GAPDH inhibition (Fig. 6d), cariporide completely rescued neutrophil death and NET formation caused by GAPDH inhibition (Fig. 6c Movies 27,28,31,32). Therefore, GAPDH inhibition kills neutrophils and causes NET formation via an increase in pH. HA-induced pH changes were partially rescued by 2DG treatment (Fig. 6b). This is consistent with the idea that pH changes caused by GAPDH inhibition are not due to a general inhibition of glycolysis but specific to a block at the level of GAPDH and in part due to accumulation of upstream glycolytic intermediates. The neutrophil pH was not elevated by treatment with methylglyoxal (Fig. S9b) in agreement with our data that methylglyoxal-induced block of the respiratory burst and pH-induced NET formation are independent events downstream of GAPDH inhibition. Treatment with the NET-inducers PMA or A23187 also did not affect the pH, indicating changes in cellular pH are not a universal mechanism by which NETs are triggered (Fig. S9c). HEPES treatment inhibited PMA or A23187-induced NET formation (Fig. S9d, e), while blocking $Na^+/H^+$ exchange with cariporide delayed A23187-induced NET formation but not PMA-induced NET formation (Fig. S9d, e). These results agree with prior work that control of pH is important for NET formation[30,73,74] and suggest that the precise mechanisms by which the pH is controlled differ between different NET-inducing stimuli.

To understand if normalization of the pH rescues any metabolic changes caused by GAPDH inhibition we performed metabolomics after treatment with cariporide or HEPES and HA. Rescuing the pH with cariporide or HEPES did not rescue any metabolic changes caused by HA including accumulation of metabolites upstream of GAPDH, depletion of metabolites downstream of GAPDH and changes in redox and ATP levels (Figs. 6e–i, S10a–i). Stable isotope tracing with $U^{13}C$-glucose or $1,2^{13}C$-glucose (Fig. S10j–o) similarly showed that rescuing the increase in pH did not rescue the effects of GAPDH inhibition on glucose flux through glycolysis and the PPP. Hence, the effects of pH on GAPDH inhibition-induced NET formation are downstream, not upstream of changes in glycolysis. These results suggest that increased pH downstream of GAPDH inhibition causes NET formation via non-metabolic mechanisms.

## GAPDH inhibition-induced NET formation is mediated by neutrophil elastase

Increased pH is known to promote neutrophil elastase activity[75–77]. GAPDH inhibition increases nuclear staining of neutrophil elastase (Fig. 2h–i), which is required for NET formation in response to extracellular stimuli[78]. We tested if neutrophil elastase inhibitors, which are currently in clinical trials as treatments for inflammatory conditions, could rescue HA-induced NET formation. Neutrophil elastase inhibitors AZD9668 or BAY-678 or GW311616a did not block HA-induced cell membrane permeabilization but blocked the extrusion of DNA outside of the cell membrane at the final stage of HA-induced NET formation (Figs. 6j–l, S11a–c, Movies 33–40). AZD9668 or BAY-678 had a similar effect on PMA-induced NET formation (Figs. 6j, S11a–b, Movies 41–44). Therefore, DNA extrusion in NET formation is not a passive consequence of nuclear expansion or cell membrane permeabilization but a neutrophil elastase-dependent process at the endpoint of diverse NET-inducing mechanisms, including GAPDH inhibition.

## Discussion

Extensive metabolomics analysis in COVID-19 patient plasma or serum has shown that disease progression is associated with changes in the levels of many metabolites[79–92]. A previous study also measured levels of 8 metabolites in neutrophils from 5–9 COVID-19 patients or healthy controls[38]. The metabolic composition of immune cell types has not been comprehensively analyzed. Our metabolomics analysis profiling almost 300 metabolites from 56 patients and 19 controls provides a roadmap for understanding the role of neutrophil metabolism in severe COVID-19, in which neutrophil dysregulation is a major pathogenic mechanism. Neutrophils from severe COVID-19 patients had alterations in the levels of many metabolites not previously implicated in neutrophil function or inflammation, suggesting new hypotheses for investigating metabolic mechanisms in neutrophils. It remains to be

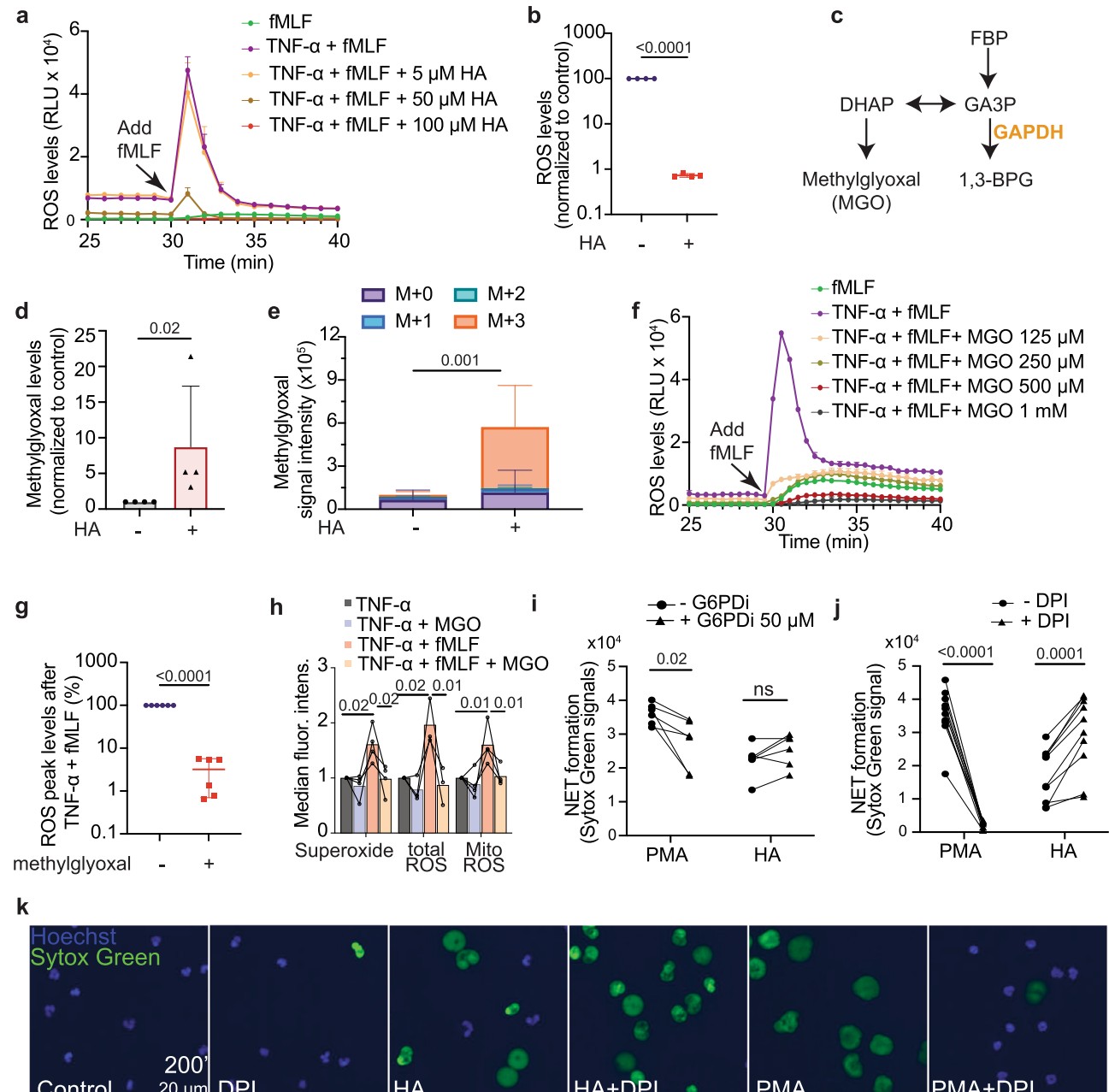

**Fig. 4 | GAPDH inhibition inhibits the neutrophil respiratory burst and causes NOX-independent NET formation. a, b** Dose-dependent effect of HA on TNF-α + fMLF-induced ROS production in the respiratory burst as measured by lucigenin signal intensity in primary human neutrophils. A representative trace from five independent experiments is shown in (**a**) and quantitation of four samples/treatment from two independent experiments using 100 µM HA shown in (**b**). ROS reactive oxygen species. **c** Schematic of methylglyoxal (MGO) production from DHAP. **d** Methylglyoxal levels in primary human neutrophils after HA treatment. n = 4 donor samples/treatment in four independent experiments. **e** Methylglyoxal labeling from U[13]C-glucose tracing. n = 7 samples/treatment from five donors in five independent experiments. **f, g** Dose-dependent effect of methylglyoxal on TNF-α + fMLF-induced ROS production in the respiratory burst. A representative trace (**f**) and quantitation of independent experiments using 100 µm methylglyoxal (**g**) are shown. n = 6 samples/treatment from three donors in three independent experiments. **h** Effect of methylglyoxal on TNF-α + fMLF-induced ROS production in the respiratory burst as measured by ROS sensors using flow cytometry from

n = 3–4 independent experiments. **i** Effect of a G6PD inhibitor (G6PDi) on PMA-induced or HA-induced NET formation assessed with a Sytox Green microplate assay. n = 6 samples/treatment from three donors in three independent experiments. **j** Effect of NOX inhibition by DPI on PMA-induced or HA-induced NET formation assessed with a Sytox Green microplate assay. n = 10 samples/treatment from five donors in five independent experiments. **k** Effect of NOX inhibition by DPI on PMA-induced or HA-induced NET formation assessed by time-lapse microscopy. Shown is one frame/condition from time-lapse movie. Time-lapse frames are shown in the Supplementary Figs. n = 3 independent experiments. Data represent mean ± st.dev. Statistical significance was assessed with two-tailed Welch's t-test on log-transformed values (**b, g**), two-tailed paired t-test of log-transformed values (**d**), two-tailed Welch's t-test (for m + 3 in panel **e**), one-way ANOVA of log-transformed values with Holm–Sidak multiple comparisons correction (**h**), two-tailed paired t-test (**i, j**). For some of the experiments in panels **i** and **j**, G6PDi and DPI treatments were tested in parallel with the same PMA or HA treatments.

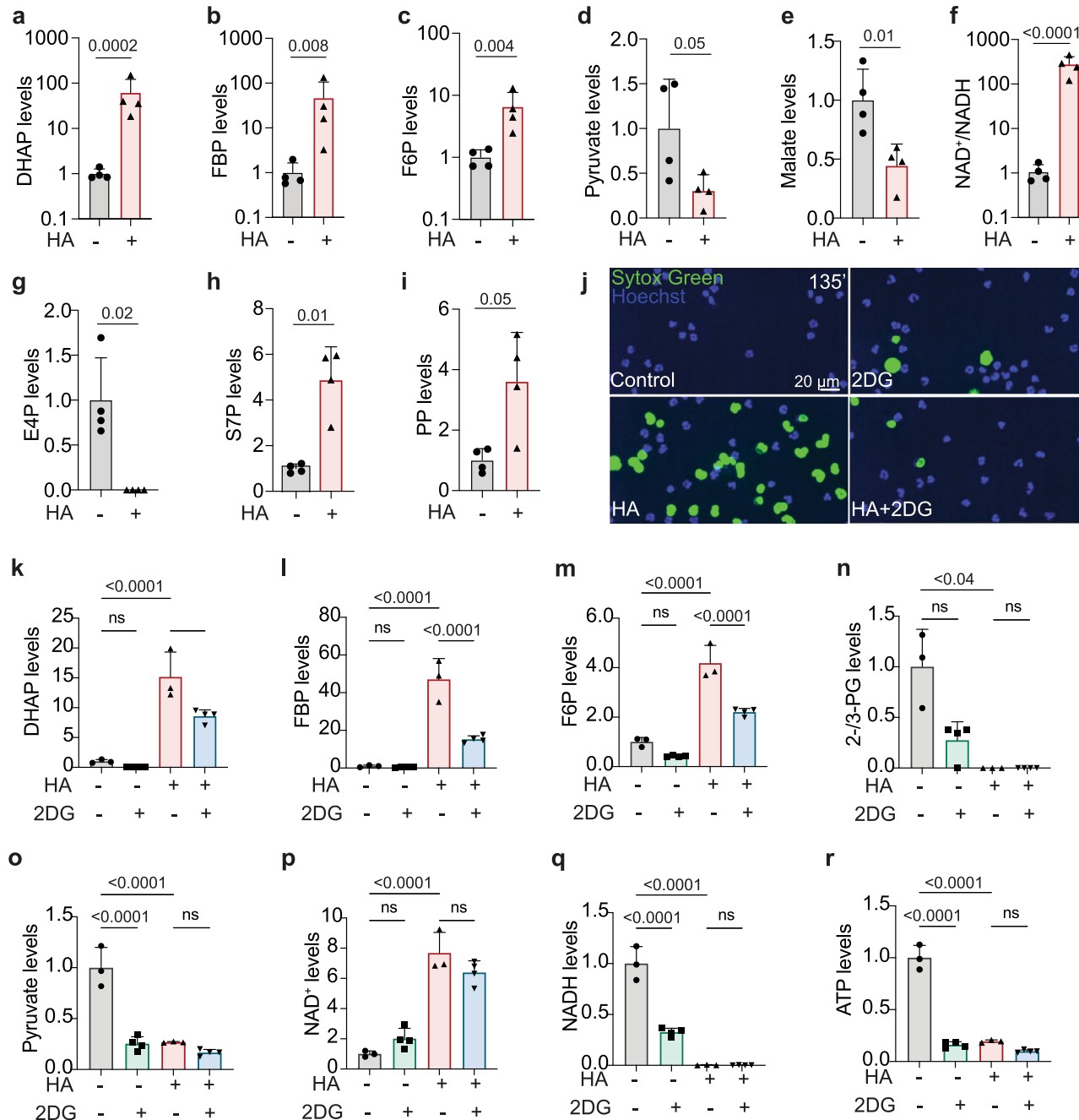

**Fig. 5 | Metabolic flow through GAPDH is a NET-suppressor mechanism.**
**a–i** Metabolite levels, normalized to controls, in primary human neutrophils 1 h after HA treatment. $n = 4$ experiments from 4 donors. **j** The effects of 2DG treatment on HA-induced NET formation as assessed by time-lapse microscopy. Magnification of one frame is shown from $n = 3$ independent experiments from three donors. **k–r** Metabolite levels in neutrophils after 2DG and/or HA treatments. $n = 4$ experiments from two donors. Data represent mean ± st.dev. Statistical significance was assessed with two-tailed $t$-test on log-transformed values (**a, b, c, f**), two-tailed $t$-test (**d, e**), two-tailed Welch's $t$-test (**g–i**), one-way ANOVA (**k–m, o–r**), Kruskal–Wallis test (**n**). $p < 0.0001$ (k-HA vs control, **l, m, o–r**), $p = 0.0062$ (HA + 2 DG vs HA), $p = 0.035$ (**n**).

tested how the metabolic changes we have observed affect the function of neutrophils in COVID-19 or other diseases. For example, the association of the histidine/β-alanine/anserine pathway with COVID-19 severity in neutrophils (Figs. 1f, g, S2c) and the association of systemic histidine depletion with lung disease and overall mortality[93] make this pathway an interesting target for future investigations of neutrophil metabolism in respiratory disease.

Many extracellular stimuli can trigger NET formation in COVID-19 or in other infections or inflammatory conditions. But it is not known to what extent molecular mechanisms actively suppress NET formation in healthy unstimulated neutrophils. Our data suggest that GAPDH

activity is required to suppress NET formation. This is likely a distinct pathway from PFKL-mediated suppression of PMA-induced NET formation, which operates by redirecting flow from the oxPPP to glycolysis[37] because HA-induced NET formation is not blocked by G6PD or NOX inhibitors. The fact that two GAPDH inhibitors, HA and iodoacetate, cause NET formation with identical kinetics and morphological changes, at concentrations sufficient to block GAPDH but not at lower concentrations, and that HA-induced NET formation is rescued by 2DG, an inhibitor of glycolysis upstream of GAPDH, or by normalization of the pH, suggest the effects of HA treatment are due to inhibition of GAPDH rather than off-target effects. Future studies,

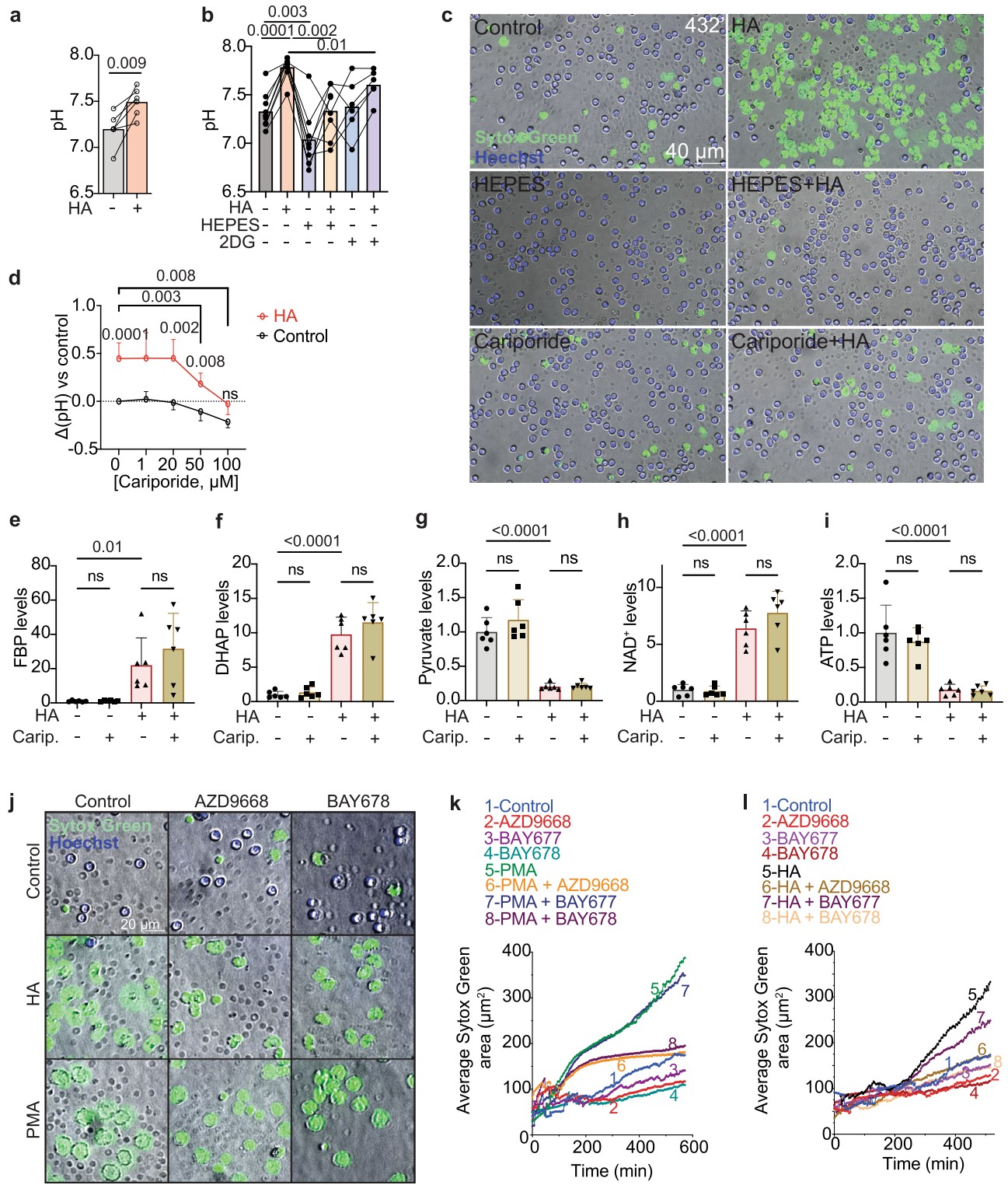

including genetic inactivation of GAPDH, are needed to address the role of GAPDH in NET formation in vivo. The neutrophil death triggered by GAPDH inhibition involves NET formation as opposed to non-specific toxic processes because it is characterized by nuclear enlargement, nuclear localization of neutrophil elastase, and DNA extrusion which can be rescued by neutrophil elastase inhibitors. Our results suggest that metabolic flow through GAPDH is a cell-intrinsic NET suppressor which prevents NET formation in unstimulated neutrophils. Thus, NET formation can be triggered not only by activation of external NET promoter pathways, but also by inactivation of an intrinsic NET-suppressor pathway. Because of the involvement of NETs

in a variety of diseases, our results raise the question of whether control of NET formation by GAPDH plays a general role in inflammation or infection.

GAPDH inhibition causes many metabolic changes. It is therefore surprising that normalization of the pH is sufficient to rescue cell death and NET formation caused by GAPDH inhibition. Normalizing the pH does not normalize other metabolic changes caused by GAPDH inhibition suggesting the increase in pH is a consequence rather than a cause of metabolic changes. Increased pH can directly activate signaling pathways that cause cell death[94,95]. It will be interesting to test if GAPDH controls the survival of other cell types via regulation of pH.

**Fig. 6 | GAPDH inhibition-induced NET formation depends on changes in pH and on neutrophil elastase. a** Effect of GAPDH inhibition on neutrophil pH, $n = 4$ donor samples/treatment from four experiments. **b** Effect of 20 mM HEPES ($n = 8$ donor samples from seven independent experiments) or 10 mM 2DG ($n = 6$ donor samples from six independent experiments) on pH in HA-treated and control neutrophils. **c** Effect of 20 mM HEPES or 100 μM cariporide on HA-induced NET formation assessed by time-lapse microscopy. Representative images from $n = 3$ independent experiments. **d** Effect of cariporide on pH in HA-treated and control neutrophils. Δ(pH) represents change from pH of untreated neutrophils. $n = 3–8$ donor samples/treatment from seven independent experiments. **e–i** Metabolite levels, normalized to controls, in primary human neutrophils 1 h after cariporide and/or HA treatment. $n = 6$ samples from three donors. **j–l** Effect of neutrophil elastase inhibitors AZD9668 (5 μM) and BAY-678 (10 μM) on HA-induced or PMA-induced NET formation. Shown in (**j**) is one frame/treatment from time-lapse movie. Additional frames are shown in Fig. S11. Quantification shows the average Sytox Green-positive area for each nucleus as a function of time after PMA treatment (**k**) or HA treatment (**l**). PMA or HA induced an expansion of Sytox Green-positive nuclei which was blocked by neutrophil elastase inhibitors. BAY-677 (10 μM) is the inactive (negative control) version of BAY-678. $n = 2–4$ independent experiments. Data represent mean ± st.dev. Statistical significance was assessed with a two-tailed paired $t$-test (**a**), repeated-measures one-way ANOVA with the Geisser-Greenhouse correction (**b**), two-tailed $t$-test (d-HA-treated vs HA-untreated, *), repeated-measures mixed-model (d-cariporide treated vs cariporide-untreated, #), or one-way ANOVA (**e–i**). $p = 0.013$ (**e**), $p < 0.0001$ (**f–i**).

It may appear paradoxical that suppression of the respiratory burst accompanied by increased NET formation is observed in COVID-19[11,58,96,97] and in other inflammatory conditions including ME/CFS[98]. Our findings that GAPDH inhibition impairs the respiratory burst and promotes NET formation via distinct mechanisms offer one explanation for this paradox. GAPDH is exquisitely sensitive to oxidant species[99,100] including nitrosylation[101–103]. Excess nitrosative and other oxidant stress may be a common pathogenic mechanism between COVID-19 and ME/CFS[104–106]. The plasma of severe COVID-19 patients is also characterized by highly elevated calprotectin, a heterodimer of S100A8 + S100A9 alarmins produced by neutrophils[5,107] which can facilitate nitrosylation of GAPDH[108]. The identification of GAPDH as an innate NET-suppressor mechanism suggests that preservation of GAPDH activity may be important to prevent NET formation in hyperinflammatory conditions characterized by oxidative stress.

## Methods

### Neutrophil samples

COVID-19 patient samples were obtained through the University of Texas Southwestern Medical Center SARS-CoV-2 Biorepository. Sample collection from COVID-19 patients or healthy controls was performed according to procedures approved by the Institutional Review Board of the University of Texas Southwestern Medical Center (STU-2020-0375 for SARS-CoV-2 Biorepository and STU 012014-040 to J. Moreland) and the UT Southwestern Biorepository. Samples from healthy donors were obtained with informed consent. For deidentified patient samples informed consent was waived. No identifiable or protected health information formed part of this study. Donor characteristics are summarized in Fig. S1b. Patients in the 'mild' category are defined as 'mild or moderate' and patients in the 'severe' category are defined as 'critical' according to WHO severity definitions (https://www.who.int/publications/i/item/WHO-2019-nCoV-clinical-2021-2). Acid citrate dextrose-anticoagulated blood was centrifuged to collect plasma. Neutrophils were isolated using Ficoll-paque density gradient centrifugation, dextran sedimentation and erythrocyte hypotonic lysis following standard procedures as previously described[109]. The number of neutrophils in each sample was counted using a microscope before freezing. Samples were stored at −80 °C until metabolite extraction.

### Neutrophil metabolomics

Metabolites from pellets containing 2 million neutrophils were extracted with 100 μl of chilled solvent of 40:40:20 methanol:acetonitrile:water (v:v:v) per million cells. The cell pellet was resuspended in the extraction solvent, vortexed for 30", and freeze-thawed three times. Ten microliters of the extract was immediately transferred to a new tube for ascorbate measurements (described below). The remaining extract was centrifuged at $21,000 \times g$ for 15 min at 4 °C and the supernatant transferred to LC-MS autosampler vials. Liquid chromatography was performed using a Millipore ZIC-pHILIC column (5 mm, 2.1 × 150 mm) with a binary solvent system of 10 mM ammonium acetate in water, pH 9.8 (solvent A) and acetonitrile (solvent B) with a constant flow rate of 0.25 mL/min. The column was equilibrated with 90% solvent B. The liquid chromatography gradient was: 0–15 min linear ramp from 90% B to 30% B; 15–18 min isocratic flow of 30% B; 18–19 min linear ramp from 30% B to 90% B; 19–27 column regeneration with isocratic flow of 90% B. Mass spectrometry was performed with a Thermo Scientific QExactive HF-X hybrid quadrupole orbitrap high-resolution mass spectrometer (HRMS) coupled to a Vanquish UHPLC as previously described[110,111]. Retention time information obtained by running chemical standards was used for metabolite identification[110,111]. A pooled quality control (QC) sample was used through the LC-MS run interspersed with the neutrophil samples to monitor the quality and consistency of detection. All samples were randomized to avoid artifactual changes caused by systematic drift.

### Ascorbate measurement

For neutrophil ascorbate measurement, 10 μl of the extraction solvent (80% methanol containing 10 mM EDTA and 50 pmoles [13]C-ascorbate/200,000 cells as an internal standard) was added to 10 μL neutrophil extract, vortexed for 10", centrifuged at $21,000 \times g$ for 15 min at 4 °C, the supernatant was diluted using 30 μL of 0.03% formic acid in water and analyzed using an AB Sciex 6500+ QTrap mass spectrometer coupled to a Shimadzu UHPLC system operating in negative mode to detect the transitions 175/115 (ascorbate) and 176/116 ([13]C-ascorbate). LC conditions were as previously described[112]. For plasma ascorbate measurement, the extraction solvent was 80% methanol with 5 mM EDTA and 5 μM [13]C-ascorbate as an internal standard. 10 μL plasma were mixed with 40 μL chilled extraction solvent, vortexed for 30", centrifuged at $21,000 \times g$ for 15 min at 4 °C and supernatant was transferred into a new tube. Ten microliters of the extract was then mixed with 20 μL of 5 mM EDTA in water, centrifuged at $21,000 \times g$ for 15 min at 4 °C and transferred to LC-MS autosampler vials. Ascorbate measurements for plasma were performed in the same way as for neutrophil samples.

### Metabolomics after ex vivo neutrophil treatments

Freshly isolated neutrophils from healthy donors were incubated at 37 °C at a density of 1 million cells/ml, counted under the microscope, in Hank's buffered salt solution (HBSS; Gibco) with $Ca^{2+}/Mg^{2+}$ and added 0.1% Glucose + 1% human serum albumin. Neutrophils were incubated with Heptelidic acid (HA) or iodoacetate (IA) at the doses and timepoints indicated in the results. Neutrophils were incubated with 10 mM 2DG, 100 μM cariporide, 20 mM HEPES, alone or with 100 μM HA as indicated in the results for 1.5 h. Samples were then centrifuged at $300 \times g$ in 4 °C for 5 min. Media were removed and cell pellets were immediately frozen in liquid nitrogen. Metabolite extraction and metabolomics analysis was performed as for patient samples.

### U[13]C-glucose and 1,2-[13]C-glucose tracing

Freshly isolated neutrophils from healthy donors were incubated at 37°C at a density of 1 million cells/ml in tracing media (Phosphate buffered saline + 0.2% U[13]C-glucose or 1,2-[13]C-glucose + 1% human serum albumin) for a total of 1.5 h. HA was added at 100 μM for 1.5 h.

TNF-α was added at a concentration of 0.1 ng/ml for the final 30 min of incubation. In tracing experiments with cariporide, cells were treated for 1.5 h. Samples were then centrifuged at 300 × $g$ in 4 °C for 5 min. Media were removed and cell pellets were immediately frozen in liquid nitrogen. Metabolite extraction and metabolomics analysis was performed as for patient samples.

## Methylglyoxal measurement

The procedure was adapted from a published protocol[113]. Neutrophils were pelleted, media was removed and metabolites extracted using 50 µl/1 million cells of 50:50 acetonitrile/methanol (v:v) + 1% formic acid + 100 µM DETAPAC. Extracts were vortexed for 30", freeze-thawed three times, centrifuged at 21,000 × $g$ for 15 min at 4 °C and the supernatant transferred to new tubes. Extracts were derivatized with 100 µM of 4-methoxy-o-phenylenediamine for 4 h at room temperature protected from light. Fifty microliters of 0.1% FA in water was added and 10 µL used for LC-MS analysis. Samples were centrifuged at 21,000 × $g$ for 15 min, and supernatants were transferred to glass vials for LC-MS/MS analysis. The mass spectrometer was an AB Sciex 6500+ operating in multiple reaction monitoring (MRM) mode. Liquid chromatography was performed using a Scherzo SM-C18 column (Imtakt). Mobile phase A was 0.1% formic acid in water and mobile phase B 0.1% formic acid in acetonitrile. The flow rate was 0.3 ml/min. The liquid chromatography gradient was: 0–2 min 0% B; 2–6 min linear ramp from 0% B to 100% B; 6–8 min isocratic flow of 100% B; 8–8.5 min linear ramp to 0% B. 8.5–11 min 0% B. A valve switch was used to allow flow into the mass spectrometer between 3 and 7.5 min. Methylglyoxal was monitored in the positive mode (175/105, 175/77, and 175/148 transitions).

## Mass spectrometry data analysis

For data acquired with the AB Sciex 6500 + , chromatogram peak areas were integrated using Multiquant (AB Sciex). For data acquired with the QExactive, chromatogram peak areas were integrated using Tracefinder (Thermo Scientific) for targeted analysis using an in-house database, and Compound Discoverer (Thermo Scientific) for untargeted analysis. The peak area for each metabolite was normalized to the median peak area of all the metabolites in each sample. Metabolomics data analysis, hierarchical clustering, volcano plots, and PCA plots were generated using MetaboAnalyst, Graphpad Prism or Origin. For metabolic pathway enrichment analysis, metabolites which differed between conditions with a false discovery rate of <0.01 were used as an input to the SMPDB metabolite database using Metaboanalyst 5.0.

## Metabolite network analysis

Linear regression was used to model the relationship between metabolite level and COVID-19 status, age, gender, sample collection time, and the interaction between COVID-19 status and these three variables (age, gender, sample collection time). Metabolite levels were adjusted by removing the effect of age, gender, and sample collection time. Autocorrelations between metabolite pairs were calculated for neutrophil samples. Metabolites pairs with Pearson correlation coefficient higher than 0.6 and correlation $p$-value <0.05 were shown as significantly associated pairs. Regression analysis and correlation calculation were performed in R. Connected metabolites were visualized as networks in Cytoscape (https://cytoscape.org).

## Live-cell imaging

Freshly isolated human neutrophils were incubated in a 96-well black imaging plate (ibidi) at a density of 75,000 cells/150 µl of media consisting of HBSS (Gibco) supplemented with Ca²⁺/Mg²⁺, 0.1% Glucose, 1% human serum albumin, 10 µg/ml Hoechst 33342, and 5 µM Sytox Green. Cells were allowed to settle for 20 min, and drugs were added at the following final concentrations: 15 nM PMA (Cayman Chemicals),

100 µM HA (Cayman Chemical or Santa Cruz Biotechnology), 5 µM A23187 (Millipore/Sigma), 5 µM AZD9668 (Cayman Chemicals), 10 µM BAY-677 (Sigma), 10 µM BAY-678 (Sigma), 5 µM GW311616a (Cayman Chemicals), 5 mM sodium pyruvate (Sigma), 5 mM sodium lactate (Sigma), 10 µM diphenyleneiodonium (DPI, Sigma), 500 µM iodoacetate (Sigma), 10 mM 2-deoxy-D-glucose (Sigma), 20 mM HEPES (Sigma), 1–100 µM cariporide (Sigma). Time-lapse acquisition began 30–45 min after addition of reagents using a fully motorized Nikon CSU-W1 spinning disk confocal microscope at 37 °C. Images were acquired at a magnification of ×40 with a Hamamatsu Orca-Fusion sCMOS camera. Hoechst 33342 and Sytox Green were imaged at low laser power to minimize phototoxicity and photobleaching. In some experiments a brightfield channel was also used. Images were captured every 2 min for experiments that did not include the brightfield channel and every 3–4 min for experiments that included the brightfield channel. Thirty to fifty wells were imaged over a period of 6–10 h. Time-lapse movies were analyzed with Fiji.

## Immunofluorescence

Freshly isolated human neutrophils were seeded on glass coverslips in 24-well plates in RPMI-1640 supplemented with glutamine, 10 mM HEPES, and 0.05% human serum albumin at 37 °C. Stimulation with the reagents described in the results was performed for 6 h. Cells were fixed with 2% formaldehyde and stained as previously described[114]. Antibodies used were rabbit anti-neutrophil elastase (1:200, Sigma, #481001), rabbit anti-citrullinated histone H3 (1:500, Abcam, # ab5103) and Alexa Fluor 488-Affinipure goat anti-rabbit IgG (1:200, Jackson Immuno Research, Fisher #NC0323535). Slides were imaged using a Nikon CSU-W1 spinning disk confocal microscope. Images were analyzed with Fiji.

## Measurement of ROS production in the respiratory burst

Freshly isolated neutrophils from healthy donors were incubated in duplicate wells in a white 96-well plate at a density of $10^5$ cells/well in HBSS with Ca²⁺/Mg²⁺ + 0.1% Glucose + 1% human serum albumin with 100 µM lucigenin at 37 °C and stimulated with 1 ng/ml TNF-α for 30 min followed by 1 µM fMLF. Lucigenin enhanced chemiluminescence was measured in a microplate reader (Lumistar, BMG Labtech) every 30 s for 60 min and measurements expressed as relative light units (RLUs). For HA, G6PDi, methylglyoxal, pyruvate, and lactate treatments, cells were pre-incubated with the reagents for 1 h.

## Sytox assay for NET formation

Freshly isolated neutrophils from healthy donors were incubated in duplicate wells in a black 96-well plate at a density of $10^5$ cells/well in HBSS with Ca²⁺/Mg²⁺ + 0.1% Glucose + 1% human serum albumin + 5 µM Sytox Green with drugs added in the concentrations indicated in the results. Sytox Green fluorescent intensity was measured in a microplate reader (Clariostar, BMG Labtech) every 5 min. Cells were pre-incubated with HA for 1 h before analysis. Other drugs were added immediately before analysis. Drugs were used as described in the section on live-cell imaging. G6PD inhibitor (Cayman Chemicals) was used at 10–100 µM as indicated in the text. In each independent experiment samples from one donor were used, and fluorescent intensity readings were normalized to the control condition of the same experiment.

## Flow cytometry

To measure the fraction of neutrophils in PMN samples, the following antibodies were used: anti-human CD49d-APC (clone 9F10, Biolegend), anti-human CD15-FITC (clone HI98, Biolegend). To measure ROS levels, neutrophils were first treated for 1 h with 100 µM HA in Hank's buffered salt solution (HBSS; Gibco) with Ca²⁺/Mg²⁺ and 0.1% Glucose + 1% human serum albumin. Then one of the following sensors was added at a final concentration of 5 µM for 30 min at 37 °C: MitoSox

## Article

Red (Thermo), CellRox DeepRed (Thermo), CellRox Green (Thermo), ROS-ID Total ROS Detection Reagent (Enzo) and ROS-ID Superoxide Detection Reagent (Enzo). Cells were washed and resuspended in the same media containing 4′,6-diamidino-2-phenylindole (DAPI, 1 μg/ml) or propidium iodide (1 μg/ml) for live/dead cell discrimination. In experiments involving fMLF, cells were analyzed with flow cytometry within 5 min of fMLF addition. Flow cytometry was performed with a FACSAria flow cytometer (BD Biosciences) or a BD FACSCanto (BD Biosciences). Data were analyzed with FACSDiva (BD Biosciences) or FlowJo (FlowJo LLC).

## pH measurement

For pH measurements, cells were incubated with 100 μM HA or other treatments for 1 h or for other durations as indicated in the text and pHrodo Red AM Intracellular pH Indicator (Thermo) was used according to manufacturer's instructions. Cells were kept on ice protected from light until flow cytometry analysis.

## Statistical analysis

For statistics of full metabolomics of neutrophils from patients, we used MetaboAnalyst to carry out ANOVA or two-tailed $t$-test after $\log_{10}$ transformation of the normalized data. To assess the statistical significance of a difference in means between two treatments, a $t$-test was used for data which did not significantly deviate from normality and did not have significantly unequal variances, a $t$-test with Welch's correction was used for data which did not significantly deviate from normality but had significantly unequal variances, and a Mann–Whitney test was used for data which significantly deviated from normality. To assess the statistical significance of a difference between more than two treatments, a one-way ANOVA was used for data which did not significantly deviate from normality and did not have significantly unequal variances, a Brown–Forsythe ANOVA was used for data which did not significantly deviate from normality and had significantly unequal variances, and a Kruskal–Wallis test was used for data which significantly deviated from normality. To test if data deviated from normality, we used the Shapiro–Wilk test when $n < 20$ or the D'Agostino Omnibus test when $n > 20$. To test if variance significantly differed among treatments, we used the F-test (for experiments with two treatments) or the Brown–Forsythe test (for more than two treatments). If the data did not significantly ($p < 0.01$) deviate from normality, we used a parametric test, otherwise data were log-transformed and tested for a significant deviation from normality. If log-transformed data did not significantly deviate from normality, a parametric test was used on the transformed data, otherwise a non-parametric test was used on the untransformed data. Multiple comparisons correction was performed by controlling the false discovery rate at 5% using the method of Benjamini, Krieger, and Yekutieli in multiple-tests in metabolomics experiments, or with the Holm–Sidak test for one-way ANOVA tests or $t$-tests, or with the Dunnett T3 test for one-way ANOVA tests with Brown–Forsythe correction, or with Dunn's multiple comparisons test for the Kruskal–Wallis test. $*p < 0.05$, $**p < 0.01$, $***p < 0.001$ unless noted otherwise. All statistical analysis was performed with measurements taken from distinct neutrophil samples from different human donors. All statistical tests comparing two populations were two-sided. Statistical analyses were performed with Graphpad Prism v9.0 & v9.3 unless noted otherwise.

## Reporting summary

Further information on research design is available in the Nature Portfolio Reporting Summary linked to this article.

## Data availability

The metabolomics data from COVID-19 patient or healthy control neutrophils have been deposited in Metabolomics Workbench under the DOI for this project (PR001600): https://doi.org/10.21228/ M8W70C. Metabolomics datasets analyzed in the current study are included in the supplementary datasets. Gene expression data for neutrophils were obtained as described from Monaco et al. (PMID: 30726743) or from the Bloodspot/BLUEPRINT database (PMID: 30395307). Source data are provided with this paper.

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

## Acknowledgements

M.A. is a Cancer Prevention and Research Institute of Texas (CPRIT) scholar and an American Society of Hematology faculty scholar. This work was supported by CPRIT (RR180007), the American Society of Hematology Faculty Scholar award, and grants from the Welch Foundation (I-2053-20210327), the Moody Foundation, the National Institutes of Health (R01DK125713 and R01HL161387) to M.A., and the National Institutes of Health (R01DK127037-01A1 and R21CA259771) to L.X. We thank Prashant Mishra and Hao Zhu for comments on the manuscript. We thank T. Mathews, L. Zacharias, D. Do, and the CRI Metabolomics Facility for help with metabolomics; C. Cantu and the Moody Foundation Flow Cytometry Facility for flow cytometry; D. Mundy and the Quantitative Light Microscopy Core (a Shared Resource of the Harold C. Simmons Cancer Center, supported in part by NCI Cancer Center Support Grant 1P30 CA142543-01 and by NIH grant 1S10OD028630-01 to Dr. Kate Luby-Phelps) for help with microscopy. This study was conducted using samples and data supplied by the UT Southwestern SARS-CoV-2 Biorepository. We thank the leadership and staff of the Biorepository, including N. Monson, B. Greenberg, D. Greenberg, M. Gill, D. Towler, and D. Russell for organizing this resource and staff members C. Pybus, J. Hadas, G. Adams, L. White, D. Sader, N Kinnare, the Neurology Translational Research Center, M. Huichapa, P. Pittman, M. Mann, F. Bernard-Vazquez, and M. McCreary for enrolling patients, collecting samples, and providing clinical data. The manuscript does not necessarily reflect the opinions or views of the Biorepository leadership or staff.

## Author contributions

Y.L. performed or contributed to all experiments. J.S.H. contributed to experimental design, isolated neutrophil samples, and performed microplate assays. Q.D. contributed to metabolomics analysis. X.X. and L.X. performed networking analysis and statistical adjustment for confounders. S.S.C. contributed to study conception. M.M. contributed to microscopy image analysis. J.G.M. contributed to study conception, experimental design, and data interpretation. M.A. contributed to microscopy and flow cytometry experiments and directed the study. Y.L. and M.A. conceived the study, designed the experiments, analyzed data, and wrote the manuscript.

## Competing interests

The authors declare no competing interests.
