## [Peer Review File · Nature Communications]

Neutrophil metabolomics in severe COVID-19 reveal GAPDH as an innate suppressor of neutrophil extracellular trap formationThis manuscript has been previously reviewed at another journal that is not operating a transparent peer review scheme. This document only contains reviewer comments and rebuttal letters for versions considered at *Nature Communications*.

REVIEWERS' COMMENTS

Reviewer #1 (expert in neutrophil metabolism):

Overall, whilst the COVID data remains descriptive, the authors have improved the manuscript by focusing on changes in neutrophil metabolism, largely removing the plasma data sets and extended the work detailing GAPDH regulation of NETosis independent of changes in ROS production. The mechanistic studies are not as strong with a small n number and the use of chemical inhibition as proof of a direct role for GAPDH, but have been extended to further explore metabolic regulation of NETosis. With this in mind, the authors should extend expand the discussion text to include this as a limitation of the current study and an area for future work. There remains no direct proof that links changes in GAPDH to changes in downstream effector functions in COVID neutrophils. Whilst I acknowledge this falls out-with the scope of the manuscript again a reference to this limitation in the discussion is needed.

I am not an expert in sesquiterpene lactones, and I acknowledge the authors response to the specificity of HA, however when using chemical inhibitors at relatively high concentrations, then it would seem likely that off target effects need to be taken into consideration. For definite proof of GAPDH regulation of Net formation, then either knockdown (currently not feasible in primary neutrophils) or study of transgenic mice would be required. Whilst this is clearly out of scope for the current ms, I still think the authors need to add reference to this as a limitation of the current work.

With respect to figure legends, the text still needs clarification regarding n number, for example in figure 4e the authors state n=7 samples from 5 donors and 5 independent expts – is this an n=5. This inconsistency is present throughout. In general, the authors would appear to use independent replicates, samples / treatment numbers, and donor numbers interchangeably, making it hard to interpret. It would also help if legends were expanded to enable stand-alone interpretation of the data included without having to return to the methodology or results section.

Reviewer #2 (expert in lipid metabolism):

The authors have addressed most of my previous concerns. However, the use of an untargeted metabolomics approach does not rationalize the lack of any internal standards for correcting variations in ion responses across individual samples in mass spectrometric analyses. In many untargeted metabolomics, several internal standards are added to improve quantitative accuracy and facilitate metabolite identification. At the minimum, a single internal standard should be incorporated for correction of inter-sample variations in ion responses. Although the QC plot in Supplemental Figure S1C showed that QCs samples were tightly clustered together, coefficients of variations (COVs) of individual metabolites across QC samples were not presented, and a typical lower threshold for untargeted metabolomics would require % COV of < 30% for individual metabolites to ensure that reliable metabolomics data were used for biological interpretations. It is advised that the authors at least acknowledge such limitation in their manuscript.

Reviewer #3 (expert in regulation of cytosolic pH, nutrient sensing):

In the current version of the manuscript, the reviewers have addressed my previous concerns sufficiently to support publication. In particular the focusing on neutrophil metabolism and the improved analysis of pH regulation and NET formation significantly improved the manuscript I fully support publication.

The only point that I still don't agree with the authors is their reasoning of pH changes in response to GAPDH inhibition. Inhibition of GAPDH causes several metabolic changes that are linked to pH regulation and thus, the lack of H⁺ production by an active GAPDH enzyme is not a convincing explanation for the observed pH changes. (i) changes in the levels of NADH and NAD⁺ (and hence the associated change in [H⁺]) are at most in the low millimolar range, and thus should not lead

to significant pH changes in a cytosol with large buffer capacity. (ii) Depletion of ATP (shifting the balance from $\text{ATP} \leftrightarrow \text{ADP} + \text{P}_i + \text{H}^+$) as observed upon GAPDH inhibition should even compensate the reduced H^+ production by GAPDH. Therefore, the authors need to reconsider these data and discuss them appropriately in their manuscript. Given the new data, the authors can rule out ATP production and glycolytic flux as critical determinants for pH regulation. Instead, the data are most consistent with the accumulation of upper glycolytic intermediates causing the pH change. This is surprising, but very interesting with respect to pH regulation by glycolysis.

Reviewer #4 (expert in lung inflammation):

The authors have addressed my previous concerns through providing additional data and discussion.

RESPONSE TO REVIEWERS' COMMENTS

Reviewer #1:

Overall, whilst the COVID data remains descriptive, the authors have improved the manuscript by focusing on changes in neutrophil metabolism, largely removing the plasma data sets and extended the work detailing GAPDH regulation of NETosis independent of changes in ROS production. The mechanistic studies are not as strong with a small n number and the use of chemical inhibition as proof of a direct role for GAPDH, but have been extended to further explore metabolic regulation of NETosis. With this in mind, the authors should extend expand the discussion text to include this as a limitation of the current study and an area for future work. There remains no direct proof that links changes in GAPDH to changes in downstream effector functions in COVID neutrophils. Whilst I acknowledge this falls out-with the scope of the manuscript again a reference to this limitation in the discussion is needed.

RESPONSE: Thanks for your comments. We show causative effects of GAPDH inhibition on NET formation, respiratory burst, and metabolism of neutrophils from healthy donors, but we did not test if GAPDH inhibition changes neutrophil function in disease models including COVID-19. We have added a reference in the discussion stating this: "It remains to be tested how the metabolic changes we have observed affect the function of neutrophils in COVID-19 or other disease models".

I am not an expert in sesquiterpene lactones, and I acknowledge the authors response to the specificity of HA, however when using chemical inhibitors at relatively high concentrations, then it would seem likely that off target effects need to be taken into consideration. For definite proof of GAPDH regulation of Net formation, then either knockdown (currently not feasible in primary neutrophils) or study of transgenic mice would be required. Whilst this is clearly out of scope for the current ms, I still think the authors need to add reference to this as a limitation of the current work.

RESPONSE: Thanks. We agree that genetic studies will be important and have added a sentence making this point in the discussion: "Future studies including genetic inactivation of GAPDH are needed to address the role of GAPDH in NET formation *in vivo*."

With respect to figure legends, the text still needs clarification regarding n number, for example in figure 4e the authors state n=7 samples from 5 donors and 5 independent expts – is this an n=5. This inconsistency is present throughout. In general, the authors would appear to use independent replicates, samples / treatment numbers, and donor numbers interchangeably, making it hard to interpret. It would also help if legends were expanded to enable stand-alone interpretation of the data included without having to return to the methodology or results section.

RESPONSE: We have updated the figure legends to include this information more consistently. In most experiments, 1 donor sample was used per day, assessed in the different treatment conditions. In some experiments, 2 donor samples were used in 1 day, or some treatments were not assessed in some experiments, or some donor samples were assessed in multiple

independent experiments, causing some differences between 'donor' and 'experiment' n numbers.

Reviewer #2 (expert in lipid metabolism):

The authors have addressed most of my previous concerns. However, the use of an untargeted metabolomics approach does not rationalize the lack of any internal standards for correcting variations in ion responses across individual samples in mass spectrometric analyses. In many untargeted metabolomics, several internal standards are added to improve quantitative accuracy and facilitate metabolite identification. At the minimum, a single internal standard should be incorporated for correction of inter-sample variations in ion responses. Although the QC plot in Supplemental Figure S1C showed that QCs samples were tightly clustered together, coefficients of variations (COVs) of individual metabolites across QC samples were not presented, and a typical lower threshold for untargeted metabolomics would require % COV of < 30% for individual metabolites to ensure that reliable metabolomics data were used for biological interpretations. It is advised that the authors at least acknowledge such limitation in their manuscript.

RESPONSE: Thanks for your comments. We have added information on the average CoV (12%) for the QCs. We appreciate the suggestion for using internal standards, which would increase accuracy for metabolites that have similar ionization properties as the internal standard. However, they would not necessarily improve accuracy for most other metabolites because their inter-sample variation in ion response may differ from the internal standard's. For metabolite identification, standards were used on the Orbitrap to establish metabolite retention times which were used for identification.

Reviewer #3 (expert in regulation of cytosolic pH, nutrient sensing):

In the current version of the manuscript, the reviewers have addressed my previous concerns sufficiently to support publication. In particular the focusing on neutrophil metabolism and the improved analysis of pH regulation and NET formation significantly improved the manuscript I fully support publication.

RESPONSE: Thank you for your positive comments.

The only point that I still don't agree with the authors is their reasoning of pH changes in response to GAPDH inhibition. Inhibition of GAPDH causes several metabolic changes that are linked to pH regulation and thus, the lack of H⁺ production by an active GAPDH enzyme is not a convincing explanation for the observed pH changes. (i) changes in the levels of NADH and NAD⁺ (and hence the associated change in [H⁺]) are at most in the low millimolar range, and thus should not lead to significant pH changes in a cytosol with large buffer capacity. (ii) Depletion of ATP (shifting the balance from ATP <-> ADP + Pi + H⁺) as observed upon GAPDH inhibition should even compensate the reduced H⁺ production by GAPDH. Therefore, the authors need to reconsider these data and discuss them appropriately in their manuscript. Given the new data, the authors

can rule out ATP production and glycolytic flux as critical determinants for pH regulation. Instead, the data are most consistent with the accumulation of upper glycolytic intermediates causing the pH change. This is surprising, but very interesting with respect to pH regulation by glycolysis.

RESPONSE: Thanks for your comments and the discussion. We agree and have clarified this in the manuscript: "This is consistent with the idea that pH changes caused by GAPDH inhibition are not due to a general inhibition of glycolysis but specific to a block at the level of GAPDH and in part due to accumulation of upstream glycolytic intermediates". We stated 'in part' here because 2DG partially but not completely rescued effects of GAPDH inhibition on upstream metabolites, pH and NET formation, hence we can't rule out that loss of H⁺ production contributes to the change in pH. We appreciate reviewer's point on the concentration of H⁺ as compared to the cytosolic buffering capacity, but the rate of H⁺ production by GAPDH is also important in this regard and if it is very high then its loss may not be sufficiently compensated by changes in other processes that regulate H⁺ concentrations.

Reviewer #4 (expert in lung inflammation):

The authors have addressed my previous concerns through providing additional data and discussion.

RESPONSE: Thank you for your positive comments.